# A Robot Operating System Framework for Secure UAV Communications

**DOI:** 10.3390/s21041369

**Published:** 2021-02-15

**Authors:** Hyojun Lee, Jiyoung Yoon, Min-Seong Jang, Kyung-Joon Park

**Affiliations:** 1Department of Information and Communication Engineering, Daegu Gyeongbuk Institute of Science and Technology, Daegu 42988, Korea; hj.lee@dgist.ac.kr (H.L.); ms3157@dgist.ac.kr (M.-S.J.); 2INTUSEER Inc., Daegu 41260, Korea; intuseer.jyyoon@gmail.com

**Keywords:** unmanned aerial vehicles, cyber-physical systems, network attack, security

## Abstract

To perform advanced operations with unmanned aerial vehicles (UAVs), it is crucial that components other than the existing ones such as flight controller, network devices, and ground control station (GCS) are also used. The inevitable addition of hardware and software to accomplish UAV operations may lead to security vulnerabilities through various vectors. Hence, we propose a security framework in this study to improve the security of an unmanned aerial system (UAS). The proposed framework operates in the robot operating system (ROS) and is designed to focus on several perspectives, such as overhead arising from additional security elements and security issues essential for flight missions. The UAS is operated in a nonnative and native ROS environment. The performance of the proposed framework in both environments is verified through experiments.

## 1. Introduction

Unmanned aerial vehicles (UAVs), commonly known as drones, have been recently deployed in various environments to perform numerous tasks [1,2,3,4]. One of the most representative drone services is Amazon’s Prime Air, which is currently under development [5]. Other companies are also preparing various drone projects such as storm damage evaluation, property damage assessment, and shale gas asset monitoring. The increase in the use of UAVs in various fields has resulted in growing concerns regarding the security of the unmanned aerial system (UAS). According to the “FAA Aviation Forecast 2019–2029” released by the Federal Aviation Administration along with the example described earlier, the commercial drone market is expected to triple by 2023. As the utilization of UAVs increases, there is a need to manage the security of the UAS. The need to manage the system security is not merely theoretical; it can be illustrated by real-life incidents. The 2011 drone hijacking incident, one of the most widely known cases of UAV cyberattacks, is an event in which Iranian cyber units cut off UAV communications links in the United States and diverted the UAVs into Iranian territory by manipulating the GPS (global positioning system) coordinates. Although there are many arguments about the authenticity of the case, it should be realized that cyberattacks on UAVs are possible. In addition, the most common vector in attack cases is the radio communication provided by the UAV platform.

UASs belong to a category of cyber-physical systems (CPSs). Unlike traditional embedded systems that operate individually, a CPS requires the close interaction of computing and physical systems. Ultimately, a CPS can be seen as the integration of computational, networking, and physical processes. It is a system in which information and software technologies combine with mechanical components to deliver and exchange data as well as monitor or control its subject by infrastructure such as the Internet in real time. The UAV network incorporates communications devices, computing functions, and control modules to form a single closed loop from data recognition, information exchange, decision making, to final execution, and it can be referred to as a CPS. In these CPSs, studies on system vulnerabilities, one of which is vulnerability in the network, are actively underway [6,7,8,9,10].

Robot Operating System (ROS) is the middleware for robot software development. Unlike the operating systems used in computers, ROS provides services such as hardware abstraction, low-level device control, and message delivery between processes for system operation. It is used in various robot industries and research fields due to its advantages such as active community and efficient development. In an UAS, ROS is installed and used on the UAV exterior board for advanced operations such as autonomous and clustered UAV. However, ROS lacks design for system security. Basic safety tools are provided, but these tools focus on system failure, such as time synchronization and program part accuracy; there are no measures for system attacks.

In this paper, we explain the vulnerability in an ROS-based UAV and propose a security framework to solve it. Section 3 describes the vulnerabilities in ROS-based UAVs and how attacks are planned using them. Section 4 describes the studies and tools that have been undertaken to address the problem. Section 5 describes the framework proposed for vulnerabilities in ROS. The performance and overhead of the proposed framework are shown in comparison with those of the tools described in Section 4; a low overhead security solution is proposed that can address vulnerabilities in ROS. Section 6 describes the proposed security framework with actual implementation and verification.

## 2. Background

### 2.1. Unmanned Aerial System (UAS)

UAS is a generic term used to denote the combination of a drone and a ground control station (GCS), as well as the communication system between the two. A drone refers to an aircraft that flies automatically or in semiautomatic mode without a real pilot on board. It performs its missions by controlling its altitude and position through an internal flight controller. The flight mission is performed either by transmission from the GCS or by built-in algorithms. The conventional media used for communication are RC transmitters, Bluetooth, Wi-Fi, and radio. Drones can send and receive commands and status from the GCS through these media using the MAVLink message protocol [11]. MAVLink is a light messaging protocol for onboard communication or components of drones. It can be implemented in 14 languages, including C and C++; various high-level APIs exist for interaction between other systems such as drones and ROS. The protocol can also be used by at least seven GCS software programs (e.g., QGroundControl and Mission Planner) to communicate with the drone. Figure 1 shows the MAVLink protocol message. Figure 2 shows QGroundControl, an illustrative GCS in UAS configuration. Figure 3 shows the UAV used in this paper.

### 2.2. Robot Operating System (ROS)

As mentioned earlier, ROS is the middleware for robot software development [13]. Unlike the operating systems used in computers, it provides services such as hardware abstraction, low-level device control, and message delivery between processes for system operation. For asynchronous communication in ROS, the publisher-subscriber model is adopted; the topic field is used for communication between the publisher and the subscriber. Figure 4 shows the structured model of ROS. The ROS consists of the master, publisher, and subscriber node. The master node connects the subscriber node to the publisher node that wants access to a specific topic. With the help of the master node, the connected publisher and subscriber nodes will be able to send and receive the desired data through the topic.

### 2.3. Rosbridge

Rosbridge is a package of ROS that allows us to use topics and services in ROS even if the client does not have ROS installed. This is possible because the JSON-based rosbridge protocol is used on the server with ROS installed. When a rosbridge server that communicates with WebSocket on the ROS server side is executed, it is possible to communicate with the node of the ROS server through various front-end devices such as the web browser, and the service is also available. Figure 5 describes the concept of rosbridge [14].

### 2.4. Safety Tool of ROS

To ensure the safe operation of ROS, there are several services that are provided by ROS. The ROS team is aware that because of the nature of the current system, the system can be attacked due to vulnerabilities in the network. To address this, they proposed a method which is not a direct function of ROS but a part of the configuration of the network used for ROS [15]. They suggest restricting access to the network and not disclosing the ROS master. There are two strategies for achieving this. The first method involves restricting hosts that can access the system. For example, there are ways to create isolated networks or use firewalls. The second method involves giving orders to authenticate users before allowing them access to the system. However, these methods are not implemented within ROS; rather, the role of protecting ROS is given to network settings outside ROS.

By default, message filtering is performed through these three functions [16]. First, there is a subscriber that acts as a top-level filter; it forwards messages from ROS to connected filters. Second, the time synchronization filter serves to synchronize to the same channels by referring to the time stamp in the headers of the receiving channels. The third function is the time sequencer. The time sequencer filter ensures that callbacks are made in temporal order according to the header timestamp of the message. When operating a robot system, the corresponding node may not be able to process the message on time, owing to various factors at the time the message was generated. When operating a robot that requires time-sensitive command input, the message filter allows the message to be processed sequentially. 

ROS also provides a tool to prepare for system failures. The Watchdog timer is a tool used for high reliability systems [17], and it is implemented in ROS. The Watchdog timer monitors the CPU and restores the system to normal conditions when abnormal or infinite loops occur. While ROS provides a Watchdog timer for these functions, it only provides the detection function and entrusts developers with a way to reconfigure the system.

The subsequent version of ROS, ROS2, introduced the concept of a management node, also called the lifecycle node [18]. It is designed for the enhanced control of the state of the ROS system. There are four node states: unconfigured, inactive, active, and finalized. Seven switching actions can be performed: create, configure, cleanup, activate, deactivate, shutdown, and destroy. When a switching operation is performed, it goes through six switching states: configuring, cleaning up, shutting down, activating, deactivating, and error processing. This node state transition is introduced to enhance the overall security of ROS.

Recently, ROS2 was officially released, and the biggest difference and feature of the previous version is the adoption of the Data Distribution Service (DDS) as middleware. DDS has several security requirements, including authentication, access control, and cryptographic operations [19]. This shows that security is important for mission-critical ROS environments. However, since ROS and ROS2 are incompatible with each other in a native environment, security issues still remain in systems using ROS.

## 3. Vulnerability Definition of UAV Using ROS

This section introduces the types of network attacks occurring in CPSs and the vulnerabilities that exist in the communication mechanisms of UAVs using the current version of ROS. 

### 3.1. Model of ROS-Based UAS

Advanced operations in UAVs require hardware and software capable of additional computing functions as well as flight controllers. The ability to provide additional assistance to the flight using information and computing power outside the flight controller is called offboard mode. The UAS assumed in this paper has a structure in which offboard computers are connected to the flight controller and communicate with each other. In addition, offboard computers support communication between flight controllers, external sensors, and offboard computers via ROS. Figure 6 schematizes the UAS with the aforementioned communication architecture from a CPS perspective. As with the CPS, this figure incorporates the plant, sensor, controller, and actuator to form a closed loop from data recognition, information exchange, decision making, to final execution. The red box indicates the part to which ROS is applied.

ROS is middleware for the development of robot software. This allows for the configuration of a UAS. The ROS adopted a publisher–subscriber (pub-sub) model for communication between each component that forms the robot. It is a structure in which two nodes, which exchange node information with the help of the master node, send messages through the publishing and subscribing functions as needed. ROS provides MAVROS, a MAVLink expandable communication node. This allows the UAV to receive the data needed for the flight over the ROS.

Figure 7, the system model of UAV with ROS through the aforementioned procedure. The /sensor node present in the external sensor publishes the message to the /process node in the offboard computer using /topic. /Process nodes deliver command messages for UAV control to /MAVROS node based on sensor data. /MAVROS node forwards the data to the flight controller via MAVLink.

### 3.2. Vulnerability of ROS-Based UAS

Table 1 defines the terms for each component in ROS. As explained in Section 2.4, ROS does not have fundamental security elements; hence, malicious nodes other than normal nodes can easily be connected. It is easy to break into the network; moreover, there is a possibility of masquerade attacks and false data injection. Once the *PA* (i.e., attack publisher) is able to configure communication with *S* (subscriber) about *T* (topic) through the master node, the false data, *msg_PAST_*, can be sent to *S*. If the attack is made, the flight controller will not be able to control the exact position and altitude in a given environment, thus allowing the attacker to destroy the system. 

There are three reasons why such attacks are possible. First, the master node does not check whether the node making the request is a normal node or a malicious node. This allows an attacker to gain unauthorized access to ROS. Hence, this enables attacks such as eavesdropping or masquerade.

Second, the master node does not check whether data from connected nodes through monitoring is within the acceptable range of the system. In any command, dropping data that exceeds a user-defined threshold protects the system from malicious false data injection. However, when an attack is authorized within the scope, the system cannot defend itself.

Third, the system does not guarantee the integrity of messages transmitted in ROS. If the system ensures that the data are authorized and not changed, it can protect itself against active attacks such as masquerade and injection attacks. Active attacks, unlike passive attacks that only eavesdrop on systems, hurt integrity and availability, and they directly affect the flight of UAVs in a short time. The second problem above can be covered if the data integrity check is satisfied. Checking data integrity can also protect the system against masquerade attacks.

The most effective attacks on the system in UAS operation are the active attacks that change the system, such as masquerade, injection, replay, etc. To protect the system against this, we need a solution that can solve the aforementioned problems. In addition, the method should not have large overheads and not obstruct the flight. We propose a security framework that addresses vulnerabilities in ROS-based UAS and has low overhead. 

## 4. Related Work

In this section, we discuss the studies that were conducted to ensure the security of UAVs. First, we present the studies for a ROS-based system. For each study, the method and direction of security application for authentication, authorization, and message verification areas are discussed.

Jeff Huang et al. [20] proposed ROSRV, a runtime verification framework for ROS-based robot applications. A node called ROSRV is placed under the master node. The node that needs to be registered as a publisher or subscriber node is identified and is connected to the other node. The second function then places the monitoring node between all publishers and subscribers within the ROS to drop commands or messages outside the user-specified range. The two functions satisfy the requirements of authorization and message verification. Thus, this can address the first and second vulnerabilities described in Section 3.2 at present. However, there are several reasons why it is difficult to apply this to UAVs, and these points are covered in Section 5.

Russell Toris et al. [21] proposed rosauth, an authentication service to enhance the security of the connection of nonnative clients in ROS. As mentioned, there is a package called rosbridge in ROS that allows clients to communicate synchronously and asynchronously with ROS, even if not in an ROS environment. The author proposed a method of authenticating whether the client accessing the ROS server using the message authentication code (MAC) is an authorized node. This project can solve the first vulnerability mentioned in Section 3.2. However, the nodes are verified using MAC only at the point of client connection. It does not guarantee the security for message tampering that occurs after the connection. In other words, it does not guarantee data integrity, the most important vulnerability in ROS-based UAS.

Bernhard Dieber et al. [22] treated ROS as a black box and used an authentication server (AS) to ensure communication between authorized nodes. In this approach, the publisher receives a key from the AS, encrypts, signs it with the message, and then forwards it to the subscriber. The subscriber can decode and verify whether the message has been tampered with. However, every time we send a message, we have two encryption overheads and a decryption overhead. Furthermore, RSA (Rivest-Shamir-Adleman) signatures are slow. 

Roland Dóczi et al. [23] proposed a security enhancement solution for ROS-based medical surgical robots. The author used authorization and authentication (AA) to eliminate security problems arising from ROS. They implemented an AA node for the AA function. The node receives its name and password from the connection request node, checks the DB, and passes the key if the information is the correct. The node then requests the master node to connect with the other node along with the key; the master node sends the key to AA to verify that it is a valid node. However, methods of authenticating using names and passwords can easily be overridden by attackers.

Ruffin White et al. [24] proposed SROS. SROS is a set of security enhancements for ROS, such as native TLS support for all socket transport within ROS, the use of x.509 certificates permitting chains of trust, definable namespace globbing for ROS node restrictions and permitted roles, as well as covenant user-space tooling to auto generate node key pairs, audit ROS networks, and construct/train access control policies. However, it is currently in an experimental development phase, and developers warn that it should not be considered as production-grade. Moreover, it is also not available to developers who use other languages because it is considered only for python development.

Manuel J. Fernandez et al. [25] used Elliptic Curve Digital Signature Algorithm (ECDSA)-based digital signatures to eliminate the security problem of communication between GCS and UAVs. ECDSA is a digital signature method based on elliptic curve encryption, and it can achieve the same level of security performance as RSA with smaller keys. Systems applying the method can address the most important security issues for the flight of UAVs by satisfying the part about message validation. The point of protecting the system through digital signatures in their study has a similar direction as our work. However, it has a concept of securing data from GCS and does not guarantee behavior in nonnative environments. Furthermore, as much as we deal with time-sensitive UAVs, we can compare the corresponding method with our proposed framework in terms of overhead. In Section 5.3.1, we compare ECDSA with the overhead of digital signatures based on SHA-256.

In addition, the security of UAV is also studied in other layers of communication. We discuss securing the UAV communications in the physical layer. Guangchi Zhang et al. [26] studied how to secure UAV-to-ground (U2G) and ground-to-UAV (G2U) communications by jointly optimizing the UAV’s trajectory to maximize the average secrecy rates of the U2G and G2U transmissions. Andrey V. Savkin et al. [27] studied the wireless communication security between a UAV to the ground node by the online planning of a UAV’s 3D trajectory. For that, they proposed a new navigation scheme with proven optimization and developed a model predictive control algorithm. Huici Wu et al. [28] developed an analytical framework for analyzing secrecy coverage performance and secrecy capacity performance. For that, they investigated secrecy performance in the air-to-ground wiretap system by considering the unique features of the UAV communication platform.

## 5. Proposed Method

The work was carried out in an environment with MAVROS, an extension package for UAV in ROS (see Section 3 for details). We found that the vulnerability of ROS makes the ROS-based UAS vulnerable. To solve this, we implemented security measures in the master, publisher, and subscriber nodes. This does not address all security issues in the system, but it ensures that the two security issues that are key to UAS operation are dealt with (see Section 4 for details),
Unauthorized users registering nodes on the system without permissionUnauthorized registered node infusing incorrect data and affecting drone flight.

Message transmission in the current ROS has the following procedure. There is the *S*-node that receives information about a particular topic *T*. *P*-node tries to transmit information about *T* and thus attempts to connect with the node that receives the *T* through the master node. The master node connects *P* to *S*. *P* broadcasts *msg_PST_* and delivers it to *S*. Then, *S* receives the information and performs calculations to control the UAV. The procedure unconditionally trusts the node and operates the robot. Thus, if a *PA*-node (i.e., the node that publishes the wrong data) accesses the system, the following occurs. *PA* requests a connection to the master node with a node that receives information for a specific topic *T*. The *PA* connected to the system injects the wrong data, i.e., *msg_PAST_*, into the *S* at a faster rate than *P*. *S* does not recognize that the data are incorrect and uses that data to control UAVs. The current procedure cannot determine whether the node that requests registration to the master is an authorized node. It is also not known whether the data that are being transmitted are modulated or are from an accredited node. For this reason, we propose a security framework for ROS-based UAS to implement a UAS that is safe from such intrusions. ROS with frameworks can be schematized as shown in Figure 8. Table 2 defines the terms for each component of the proposed framework.

### 5.1. Registration of a New Node

Access control is the function of allowing or denying someone the use of a resource. We apply access control to ROS, thus preventing unauthorized system registration of nodes. 

*ACT* means a list of access rights for nodes accessing a particular topic *T*. This includes nodes with access to *T* and can be expressed as *ACT =* [*x, y, z*]. Here, *d(P)* and *d(S)* mean digests for *P* and *S* respectively, and *d(P)* can be expressed in *H(k, P_name_||T)*. The ROS with access control registers the node using the following procedure: All publishers and subscribers accessing a specific topic T before ROS operation are listed and recorded in *ACT*. The information recorded in *ACT* is *d(P)* and *d(S)*, which are digests of *P* and *S*. The reason for recording the digest is to make it impossible for an attacker to masquerade itself as a node that sees the digest and has authority over *T*.*P* requests the master node to register *P* as a publisher of *T*.The master node obtains *d(P)*, which is the digest for *P*.The master checks whether the digest is in *ACT*. If there is digest in *ACT*, *P* is allowed to publish to *T*.

Figure 8 is a diagram showing the registration procedures of the ROS with added access control.

### 5.2. Signature with HMAC

A digital signature is a security tool that uses encryption for data integrity, authentication and denial prevention. Generally, when key sharing is not possible, we use a digital signature using RSA or ECDSA. This means that when a message is signed and sent by the private key, the receiver verifies the message with the public key. In addition to RSA, there is also a signature method using the hash-based message authentication code (HMAC). 

If the network is in an environment that does not require key exchange, HMAC has many advantages over using RSA and ECDSA. First of all, it is very fast to sign and verify, and it is simple to implement. It also has advantages in safety issues. A characteristic of ECDSA is that for the hash function used, it must be collision-resistant, but the HMAC does not have such a characteristic. Given these characteristics, digital signatures using HMAC are more advantageous than RSA and ECDSA in networks operating UAS. Therefore, we ensure data integrity by using HMAC to sign messages sent from ROS.

*H(k, msg)* obtains digests for the message (*msg*) using the key *k*. At this time, *k* should be exchanged between transceivers in advance. There are several types of hash functions, but SHA-256 was used in the solution. The SHA-256 algorithm used in the experiments is impossible to break while operating the UAS. Even if an attacker takes the time to find out what the hash is, it is also impossible for the actual system to fail because of the short cycle in which the key *k* is exchanged. The hash function can be used as the agreed function between the sender and receiver. Here, *a||b* means the connection between the letters *a* and *b*. For example, the result of *a||b* is “*ab*”.

The subscriber and publisher of ROS, where the verification process has been added, sends and receives data through the following procedure:*S* and *P* for a specific topic *T* make a registration request to the Master node.P performs a signature on *msg_PST_*. *Sign(k, msg_PST_)*, the signature procedure, means *H(k, msg_PST_)||msg_PST_*.P, which carried out the signature, sends *s = Sign(k, msg_PST_)* to *S*.*S* performs a *Verify(s)* on the received *s*.Separate *s* = *H(k, msg_PST_)*||*msg_PST_* from *H(k, msg_PST_)* and *msg_PST_*.For separated *msg_PST_*, perform *H(k, msg_PST_)* using a preshared key *k*, where *k* is the same symmetric key used by *P*.Compare the two *H(k, msg_PST_)* and inspect them for the same value.If there is no problem with *msg_PST_*, use that data.

Figure 9 is a diagram showing the data transmission procedures of the ROS with the added verification.

### 5.3. Performance and Conceptual Comparison 

#### 5.3.1. Encryption Overhead

The proposed security framework has additional features to address vulnerabilities in the existing ROS. This function results in additional overhead for data size and computation. First, the computational overhead that occurs during access control execution occurs once upon initial connection, so it does not affect UAV operation. However, for signatures that use HMAC, data overhead and computational overhead exist for each transfer.

Previously studied [22,25] also used cryptographic methods to address vulnerabilities in ROS, and there exists overhead for them. First of all, for data overhead, existing 69-byte size geometry_msgs/PoseStamped messages have 256 bytes of overhead when using RSA-2048 signatures, as in [22]. Using 256-bit ECDSA signatures used in [25] results in 64 bytes of overhead.

The computational overhead indicates how long it takes to perform an encryption. Figure 10 is the benchmarking result using the crypto++ library, with RSA-2048 taking 2.32 ms to sign and 0.05 ms to verify. ECDSA takes 1.03 ms to sign and 0.82 ms to verify.

With HMAC digital signatures used in our proposed framework, the data overhead is 32 bytes. For computational overhead, 295 MiB per second can be processed when using 128-bit keys. This means that it takes 0.0029 ms to process 101 bytes of data that combines the original message with the data overhead. Because digital signatures using HMAC use symmetric keys, verifying takes the same amount of time as signing.

Based on these results, we can see that the proposed security framework has very little overhead compared to the existing studied systems and has little impact on the performance of the existing ROS.

#### 5.3.2. The Use of MAC

We ensured integrity through verification of the data transmitted within the system using MAC. Similarly, rosauth [21] in Section 4 wanted to use MAC to improve the security of ROS. However, its use is different from the one in this study. In previous work, MAC was used in a nonnative environment early in the connection to enable clients to authenticate themselves with the server as validated clients. However, as there is no solution for the integrity of the data transmitted, the system will be breached if an attacker attempts an MITM attack on an already connected channel. Conversely, the framework proposed in this study uses MAC to ensure data integrity and authentication with each transmission since the beginning of the connection. In addition, only nodes authorized through access control can be registered. Furthermore, the proposed framework can be secured in a nonnative environment, similar to their study. This is demonstrated in Section 6 with an experiment.

#### 5.3.3. Message Verification Performance 

In our proposed framework, we can verify the presence of abnormalities in messages using HMAC digital signatures. Similarly, the ROSRV [20] in Section 4 differs from our method. Even though the authors studied message verification, there are some limitations to this solution. First, for monitoring purposes, the monitoring node between the publisher and the subscriber verifies the message. This will take twice the transmission time in the existing publisher–subscriber model. The second is that if a large number of nodes are connected to a centralized ROSRV, there will be a delay in the monitoring node. In addition, if the data are modulated within the monitoring range, it will not be detected. The existence of two overheads and the absence of integrity make it difficult to apply ROSRV to UAS.

In our proposed method, on the other hand, the sender sends the message with a digital signature, and the receiver checks the message’s digital signature to proceed with the message verification. Therefore, there is no overhead for transmission time other than the overhead described in Section 5.3.1, and there is no bottleneck in transmitting the message.

## 6. Test

This section describes an experiment that studies the consequences that can be caused by the vulnerabilities in an existing ROS-based UAS and the impact on the UAS after applying the security framework. First, we describe the experimental environment of the drones that make up the UAS for the experiment and the arrangement of the components. The results are presented with an explanation about the operation of the proposed security framework in a native ROS environment and a nonnative ROS environment.

### 6.1. Experiment Environment of UAS

Figure 11 shows the UAS environment configured for experimentation. Pixhawk is an industry standard autopilot developed and jointly developed by 3DR Robotics and Ardupilot Group. Various robots such as RC cars, airplanes, and multicopters can be made, and firmware is provided for them using Pixhawk. We made quadcopters that belong to a class of multicopters, and we used them for the experiment. Pixhawk typically uses two firmware, i.e., Ardupilot and PX4. We used PX4 firmware that supports offboard mode in the experiment because we assumed UAS to operate advanced drones such as autonomous driving and cluster flight using offboard mode. Pixhawk uses the MAVLink protocol for communication. MAVLink is a light messaging protocol for onboard communication or components of drones. This can be implemented in 14 languages, including C and C++, and various high-level APIs exist for the interaction between other systems such as drones and ROS. The companion computer used Raspberry Pi, which is an embedded Linux-based development small computer, and Ubuntu MATE, which is a Linux-based OS, was used in the computer. In using ROS with Raspberry Pi, this setup has better compatibility on a variety of issues, such as packages and kernels. ROS stands for Robot Operating System, which is not similar to the conventional operating systems used in computers. It is a middleware concept for robot development that is installed on an OS such as Linux or Windows. We installed ROS Kinetic for the experiment. ROS supports node-to-node communication using XML-RPC and TCP. XML-RPC is an XML-based distributed system communication method that is simple and portable RPC protocol over HTTP. This is used in ROS by the publisher and the subscriber to communicate with the master node to connect with each other. When the publisher sends data for a particular topic after the connection, it serializes the data and sends the data to TCP payload. The subscriber receives the packet and receives the data by deserializing it. We use the MAVROS package, which is an ROS expansion package. This package enables MAVLink communication between Raspberry Pi and Pixhawk where ROS runs. Hence, the /mavros node, which receives data related to the flight from the publisher, forwards the data to Pixhawk through the MAVLink protocol. Upon receiving this, Pixhawk calculates flight control from the flight stack based on the corresponding data. 

The overall experimental environment is described above. We conducted the experiment in this experimental environment by considering two situations. In Figure 11, two computers and one sensor that are authorized can be found attached to the ROS through the wireless network. These devices can access the ROS in a native environment or, depending on the intention of the user, the ROS can be accessed in a nonnative environment. In these two environments, the approach to ROS is as follows: First of all, if the client is in a native ROS environment, the client has ROS installed, and by running the launch file, the client accesses the ROS server and creates a node. If the client is in a nonnative ROS environment, the client does not have ROS installed and requests the master to connect to the communication for a particular topic on the front-end implemented with the roslibjs library. After connection, the client encodes the data in JSON and sends the data to the rosbridge server. On the server side, rosbridge is run, which transmits data received by clients to nodes that subscribe to the topic. 

### 6.2. Experiment on Native ROS Attack

This section describes the modes of attacks in native ROS and the ways that can be used to defend the native ROS through the proposed framework. The attacks in the environment are shown in Figure 12. First, the accredited device sends the data and commands necessary for the flight to /mavros via a specific topic. The drone performs normal flights based on their data. At this time, a malicious computer breaks into the network with the aim of sabotaging the system and then register the publisher with the ROS that transmits the /mavros/local_position/pose topic. An attacker could then influence the flight path of the drone by means of the corresponding topic. This experiment can be found in [29,30].

The actual experiment was conducted by flying a normal drone driving at a height of 2 m while considering the drone, property, and human casualties, and by returning the drone to its starting point. Figure 13 shows the state of the UAV during an attack. The *X*- and *Y*-axis in Figure 13 denote the time and the altitude of the drone, respectively. It can be seen in the figure that up to the 30 s point, only the accredited node approaches the ROS and transmits the /mavros/local_position/force topic, resulting in a 2 m high UAV flight. After that, the attacker node can then approach the ROS and inject itself into the UAV to fly the drone at an altitude of 0 m to confirm that the altitude of the UAV is slowly converging at 0 m.

The reason for this attack is the lack of verification and data integrity for newly registered nodes. We apply a security framework to the existing ROS for it to defend itself against such attacks. Figure 14 shows how HMAC is applied to ROS to send and receive data. The framework is applied to each computer running publisher and to the computer running MAVROS. We demonstrate through experiments that UAV with these security frameworks have no impact on existing methods of attack Figure 15 shows the state of the UAV during an attack in the same scenario as the above. An attack was made near 30 s, but it can be confirmed that the UAV flies at an altitude of 2 m until the experiment is over.

### 6.3. Experiment on Nonnative ROS Attack

Rosbridge is a package that enables the synchronous and asynchronous communication of ROS in an environment where ROS is not installed. We implemented a web client that can use ROS using the roslibjs library for experiments. At this time, the server side should run rosbridge to create a node that sends data to the MAVROS. Figure 16 briefly describes the rosbridge used. For practical use of rosbridge, three applications must be run on the server side. The first is ROS, and the second is the rosbridge server. When the rosbridge server receives a message from the client regarding which topic to send, it attempts to connect with the subscriber receiving the topic. In Figure 16, the corresponding subscriber is /mavros. The third application is the web server. The web server provides roslibjs services to clients via web pages and helps them communicate indirectly with ROS through rosbridge. These three applications do not necessarily have to run on one computer, and they can also run on multiple server computers. This will allow the client without ROS to communicate with the ROS installed computers. This experiment can be found in [31,32].

The attack and defense experiments in nonnative environments, as in previous experiments, proceeded with an attack that lowered the UAV flying at certain altitudes to 0 m and a scenario that defended them. Figure 17 shows the method of attack in a nonnative ROS environment. An attacker can execute an unauthorized web server to execute a malicious node on the ROS through the rosbridge server. These nodes can break UAS by injecting incorrect data into the system, such as malicious nodes in a native ROS environment. When a user enters an altitude in the text box of a web page sent from a web server, the UAV flies to that altitude. Figure 18 shows the status of the UAV affected by the corresponding attack method. Upon receiving /mavros/local_position/pose topic data from normal web clients, the UAV flies at an altitude of 2 m during 30 s. After 30 s, an attacker uses rosbridge to connect a malicious node to the ROS and inject false data to lower the altitude of the UAV. We performed data integrity and node verification by applying the HMAC-based security framework to Web servers and MAVROS. Experiments show that existing attack methods have no impact on UAVs with that method. Figure 19 shows the state of UAV when an attack is made in the same scenario as earlier. An attack was made near 30 s, but UAV can confirm that it performs a highly normal flight of 2 m until the experiment is over.

## 7. Conclusions

With the noticeable growth in the use of UAV, the security of the system has become a major concern in recent years. Due to the absence of system security, UAVs that are applied in diverse places are exposed to potential risks. Therefore, it is necessary to be aware of this fact and study the security of the system of UAVs.

For advanced operation of UAVs, computers that can operate and communicate are required in addition to the flight controller, which is referred to as offboard systems. UAS is a generic term for controls, communications equipment, etc. to operate UAVs, and it falls under the category of CPS. We investigated the vulnerability of the UAS using offboard systems in terms of the CPS, and we proposed a security framework to address it. The framework ensures the integrity of the data transmitted in the system through digital signatures and prevents unauthorized nodes from accessing the system without authorization, hiding their identities. By measuring overhead for computations, data, and transmission speeds as the framework’s functions are added, the framework is shown to be an appropriate framework for UAVs. 

In this study, the real-time experiment shows that the UAS fails to function properly through cyberattacks that use the vulnerability of the ROS and install ROS in the offboard computer. To address this, the proposed security framework was applied to the system to demonstrate system security through practical experimentation.

In the current framework, the system was defended against attacks that inject abnormal data into UAV flight by granting only access control and integrity. As a future work, we will develop a customized module that can easily upload various functions necessary for system security into the framework.

## Figures and Tables

**Figure 1 sensors-21-01369-f001:**
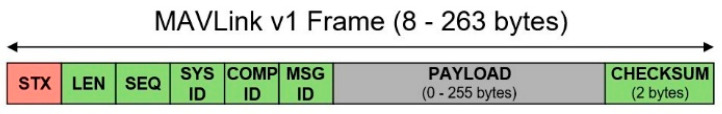
MAVLink protocol message [12].

**Figure 2 sensors-21-01369-f002:**
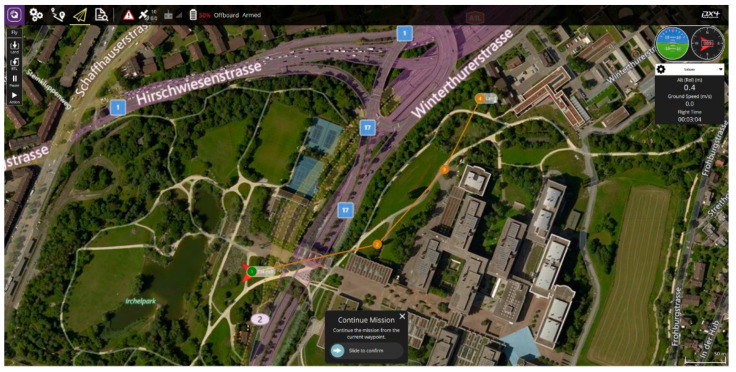
QGroundControl as the ground control station (GCS).

**Figure 3 sensors-21-01369-f003:**
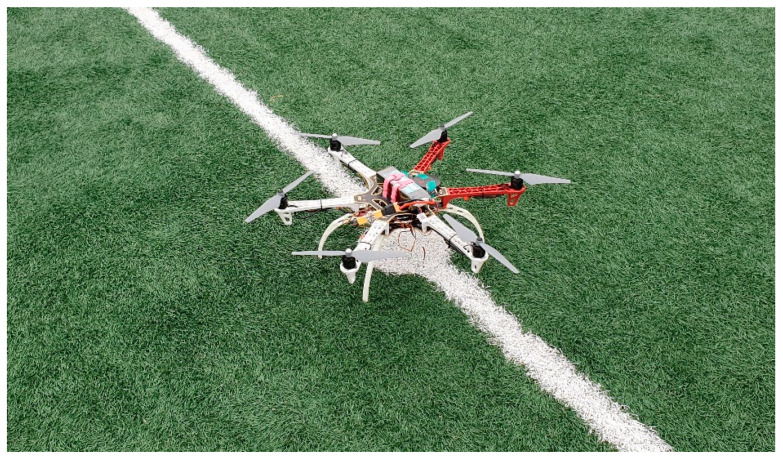
Unmanned aerial vehicle (UAV).

**Figure 4 sensors-21-01369-f004:**
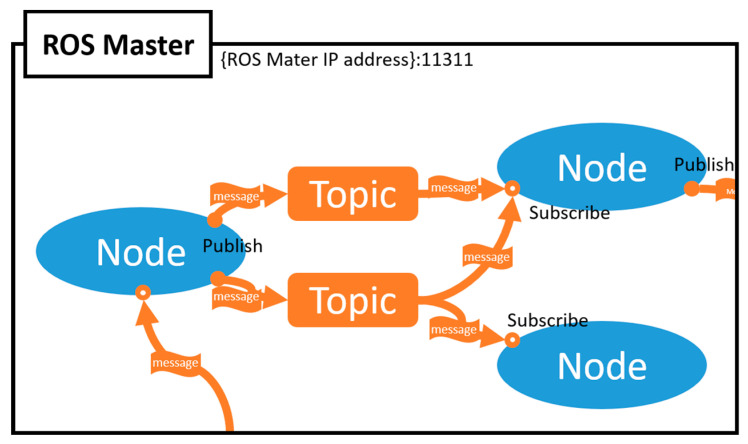
Robot operating system (ROS) structure.

**Figure 5 sensors-21-01369-f005:**
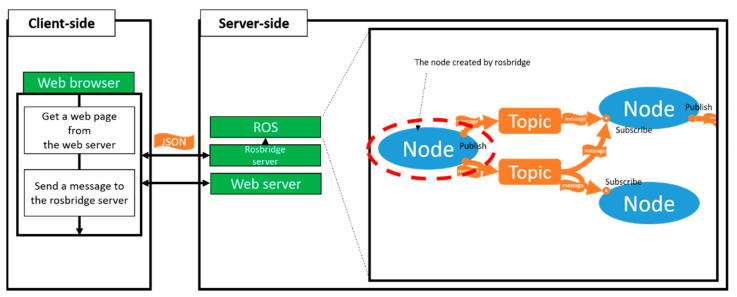
Rosbridge concept.

**Figure 6 sensors-21-01369-f006:**
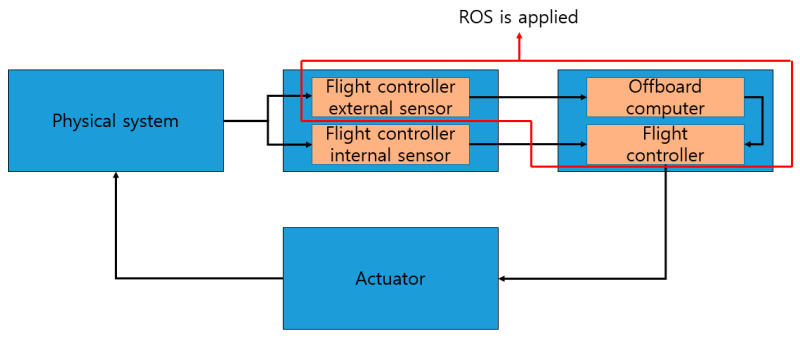
Feedback loop in an unmanned aerial system (UAS).

**Figure 7 sensors-21-01369-f007:**
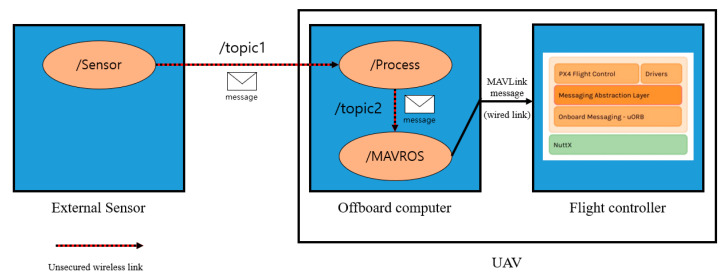
System Model of UAV with ROS.

**Figure 8 sensors-21-01369-f008:**
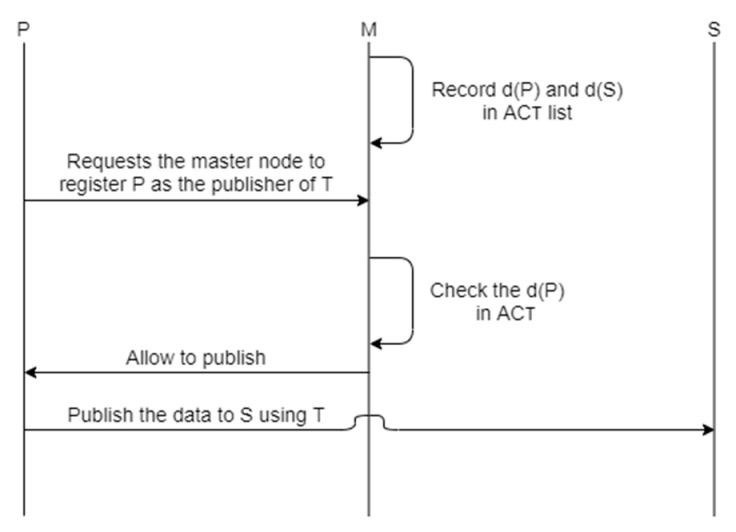
Proposed access control procedures.

**Figure 9 sensors-21-01369-f009:**
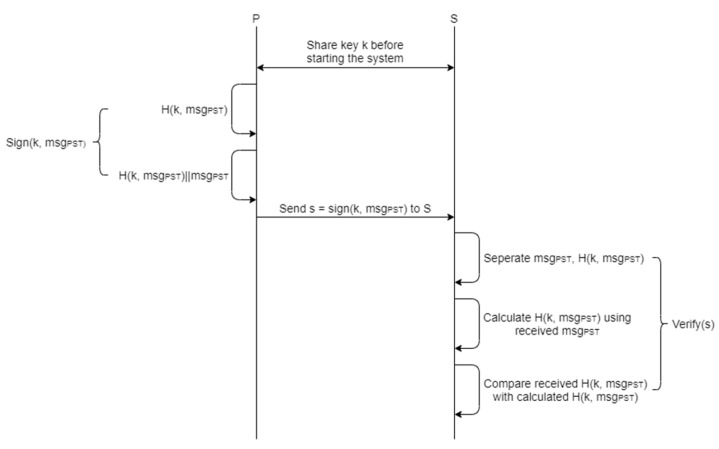
Proposed signature procedures.

**Figure 10 sensors-21-01369-f010:**
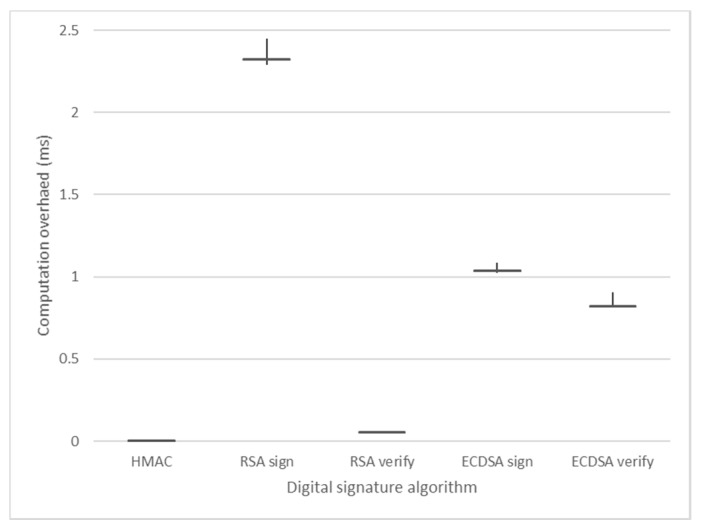
Digital signature benchmark result.

**Figure 11 sensors-21-01369-f011:**
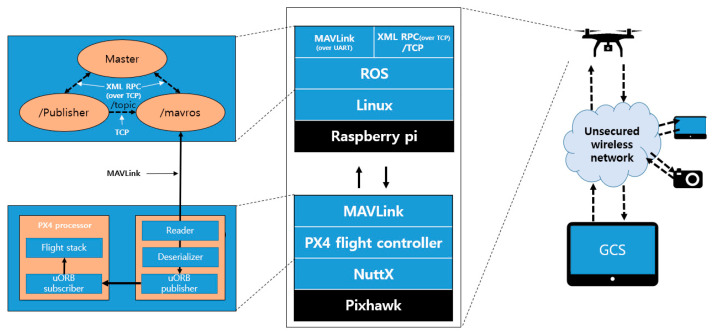
Experiment environment.

**Figure 12 sensors-21-01369-f012:**
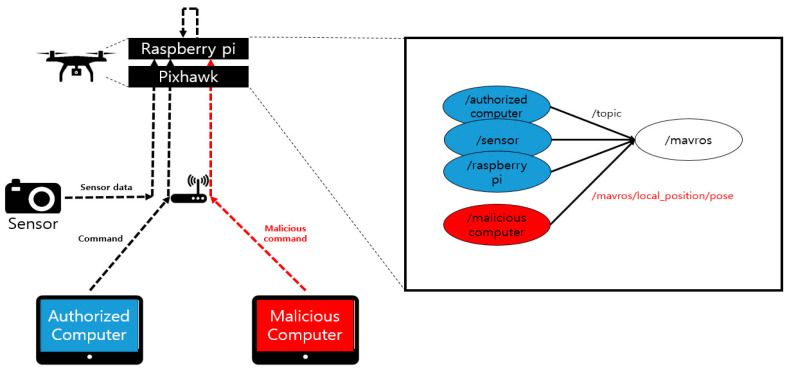
Attack in native ROS.

**Figure 13 sensors-21-01369-f013:**
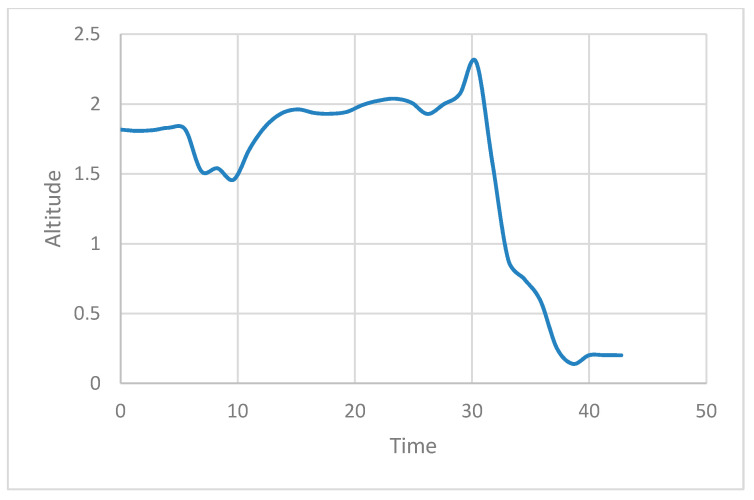
UAV flight altitude without security framework in native ROS.

**Figure 14 sensors-21-01369-f014:**
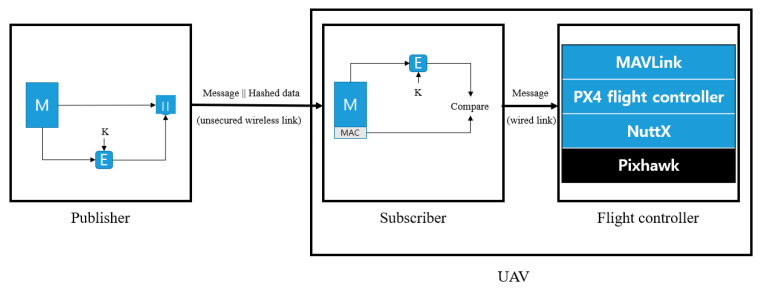
Security framework for ROS.

**Figure 15 sensors-21-01369-f015:**
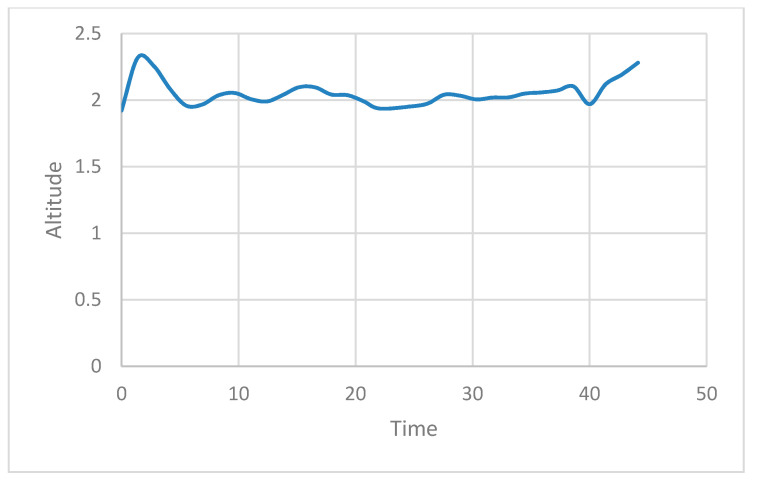
UAV flight altitude with security framework in native ROS.

**Figure 16 sensors-21-01369-f016:**
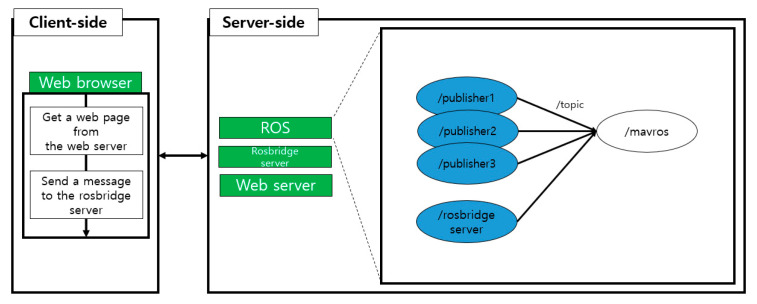
Rosbridge diagram.

**Figure 17 sensors-21-01369-f017:**
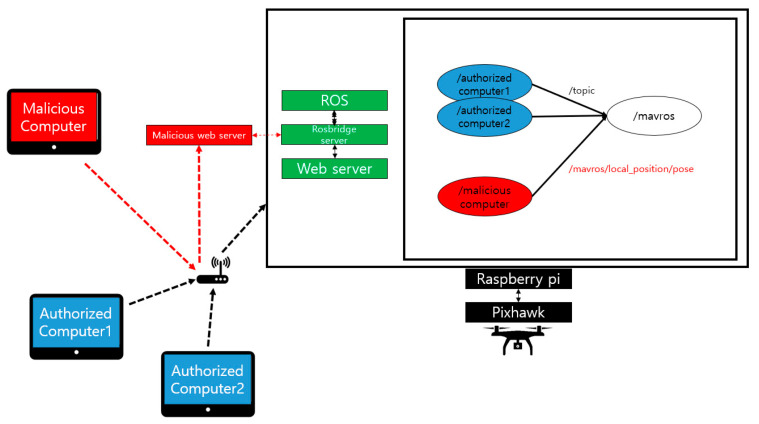
Attack in nonnative ROS.

**Figure 18 sensors-21-01369-f018:**
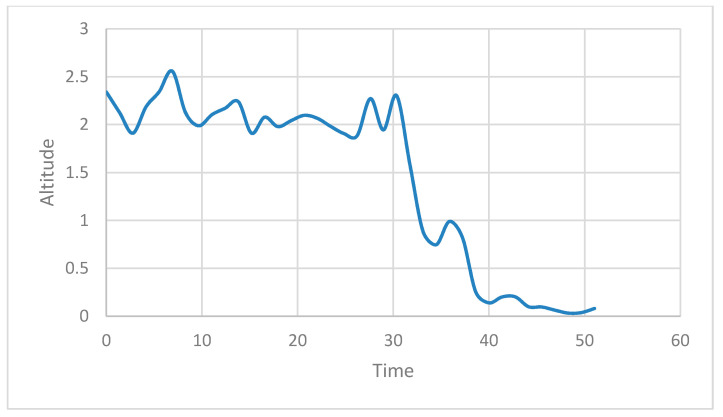
UAV flight altitude without security framework in nonnative ROS.

**Figure 19 sensors-21-01369-f019:**
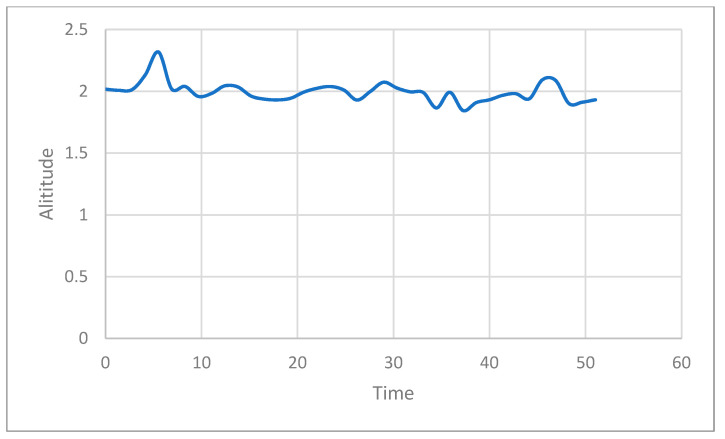
UAV flight altitude with security framework in nonnative ROS.

**Table 1 sensors-21-01369-t001:** The terms used in ROS.

Terms	Concept
*T*	Topic
*P*	Publisher to send information about a particular topic
*S*	Subscriber to receive information about a particular topic
*A*	Attacker node
*msg_PST_*	Message between *P* and *S* for *T*
*msg_PAST_*	Message between *PA* and *S* for *T*
*msg_PSAT_*	Message between *P* and *SA* for *T*

**Table 2 sensors-21-01369-t002:** The terms used in the framework.

Terms	Concept
*ACT*	Access list for *P* and *S* accessing specific topic *T*
*d(X)*	Digest for node *X*
*H(k, msg)*	Getting a hash of the message (*msg*) using a key (*k*)
*P_name_, S_name_*	Node name of *P* and *S*
*Sign(k, msg)*	Digitally signing for message *msg* using the key *k*
*Verify(s)*	Verification process for signed data s.

## Data Availability

Not applicable.

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
