# Peer review of "A Robot Operating System Framework for Secure UAV Communications"

_sensors, 2021, doi:10.3390/s21041369_

Round 1
Reviewer 1 Report
The authors work on developing relatively unexpensive Unmanned Aerial Systems (UAS) using an open-source library of software and tools (“Robot Operating System” or ROS) to speed up the development process. Yet, such applications may be exploited by hackers pursuing a variety of purposes; hence, the interest in providing cybersecurity of such systems. The authors suggest two enhancements of ROS – mandatory registration of a new node in the system and coding the messages between nodes (using hash-based message authentication code). Then they demonstrate that this is sufficient to protect the UAS against a certain type of cyberattacks (man-in-the-middle, MITM, maliciously injecting commands to the flight controller) while the computational overhead of the protection measures is ‘acceptable.’
The protection of UAVs/drones from cyberattacks is a known issue, since most UAVs are controlled remotely, at least at some points in their flight. The authors illustrate the issue with a well-known example when a rather sophisticated UAV was highjacked.
It does not matter for what purposes remote control has been introduced, since the authors discuss the provision of cybersecurity. Thus, the focus in the abstract on obstacle recognition and avoidance (assumingly, through remote control in this case) is quite confusing.
Hence, the article will benefit if the authors include a general overview of the cyber vulnerabilities of UAS (based on most recent sources, e.g., https://doi.org/10.1007/978-3-030-58703-1_8; https://doi.org/10.1016/j.aap.2020.105837; https://doi.org/10.1007/978-3-030-37393-1_18; https://doi.org/10.1109/ACCESS.2020.3037705; https://doi.org/10.1016/j.cose.2019.05.003; https://doi.org/10.1109/SSRR.2017.8088163) and then narrow down the examination to ROS-based systems (e.g., https://doi.org/10.1109/SYSCON.2019.8836824).
Some of these sources also address cybersecurity solutions. Such examination of recent sources is needed to assure that the approach presented in this article is original, at least in some aspects. (The three sources covered in section 4 are published in 2014 and 2016, which is not sufficient for this purpose.)
The second experiment is not explained properly. It is not clear whether the UAV went down from 2 to 1 m height due to the partial effectiveness of the protection or that was intended (I guess the latter, but clarification is still needed).
The article would be of interested to readers willing to test new approaches and aiming for quick fixes, e.g., university students.
Specific remarks:
Sub-title 3.2 is repeated twice on p. 7. It needs to be changed in the second occasion.
There is reference to Table 5.1, but such table does not exist.
Sources 29 and 30 use the same link to Youtube.
Some of the figures are not very informative, e.g., figure 19, while others, for example figures 6 and 7, can be combined in one.
The quality of language differs among sections and overall needs to be improved significantly. Examples:
- The authors refer to ‘stability’ (twice in the introduction, four times in the conclusion, and elsewhere) while they certainly mean ‘security’
- use of terms that have not been defined and are unclear, e.g., “active attacks” (line 285)
- 6 & figure 6 – ‘perception’ or ‘recognition’ layer
- causing the system to be destroyed [it is very easy to avoid passive voice here]
- when an attack is authorized within the scope, it cannot be defended
- if the system is satisfied with the data integrity
- passes the key if it is the correct information
- P requests the master node to master node as a publisher of …
- communication is made …
- In the actual experiment, the experiment was conducted …
- Figure 17 briefly describes …
- advanced operation of UAVs that make up UAVs
- computers that can be operated and communicated …
- operate UAVs, including UAVs;
- UAS flying offboard …
- the flight controller, which is referred to as offboard systems, etc.
Author Response
Please see attached PDF file with this response.
1. We deleted the words that cause confusion in Abstract: obstacle recognition and avoidance.
2. Several systems mentioned in Section 4 are covered in Section 5.3. New additions to Section 4 were also included in Section 5.3 and compared.
3. The second experimental graph showed Rosbridge characteristics by lowering its altitude using a regular client, but was changed to a constantly hovering experimental graph at 2 m to avoid misunderstanding.
+ We modified ambiguous and unnatural expressions.

Reviewer 2 Report
The goal is to add security to the communications in a ROS-based UAS system. The topic is timely, but it has not been properly addressed.
In general, wording is quite improvable.
The background section has not been properly structured: there are subsections with the same title, other subsections composed of just one or two paragraphs...
Section 3 is verbose, with a lot of unnecessary (and redundant) information about CPS and ROS. Only 3.4 goes to the point. Again, 3.2 and 3.3 share the title.
In Section 4 authors mention several related works, but they do not compare later their proposal with any of them.
The security mechanisms proposed in Section 5 (registration and encryption) are extremely basic.
The evaluation is not valid. As indicated, it should have considered state-of-the-art security techniques for this environment.
Overall, I consider that the manuscript does not deserve publication. There are too many (presentation and technical) deficiencies.
Author Response
Please see attached PDF file with this response.
1. We modified the configuration of the background section. Unnecessary subsections were combined and included in the parent section.
2. Unnecessary information has been removed in Section 3. We removed the existing 3.1 and 3.2 and replaced 3.3-Model of ROS-based UAS and 3.4-Vulnerability of ROS-based UAS.
3. Several systems mentioned in Section 4 are covered in Section 5.3. New additions to Section 4 were also included in Section 5.3 and compared.
4. The proposed security mechanisms in Section 5 are simple. Instead, it has less overhead than previously studied, making it suitable for time-sensitive UASs. This is referred to in Section 5.3.
+ We modified ambiguous and unnatural expressions.

Reviewer 3 Report
- In Section 2.4.1 related to network security, the authors mention that “the role of protecting ROS is given to network settings outside ROS”. It would be useful to further motivate this setting, e.g., in order to keep the ROS implementation lightweight, etc., as well as the potential implications that arise when vulnerabilities are addressed outside the ROS itself.
- In related work, relevant papers employing elliptic curve integrated encryption schemes to improve UAV security should be discussed. The authors should also mention the recent efforts of the open source robotics foundation to introduce secure ROS (sROS) and address security limitations of ROS in terms of encryption, authentication, etc.
- The key contribution of the authors is to devise a security framework with low-overhead that addresses vulnerabilities in ROS-based UAS. A summary paragraph where this contribution is differentiated with respect to the related work discussed in Section 4, would be useful to better understand the novelty beyond the state of the art.
- The authors should comment on the capability of their proposed method to address denial of service attacks on specific ROS nodes, e.g., publishing a large number of falsified data in ROS.
- Please comment on the altitude difference between Figure 19 (native ROS) and Figure 21 (non-native ROS), after the attack has been executed.
- A comparative approach in Section 6 with respect to other benchmark security enhancement schemes addressing these types of attacks, e.g., rosauth [25], would reveal more clearly the novelties of the proposed method. If the implementation of these methods is not feasible in the experimental setup, the authors should comment on the qualitative advantages of their approach with respect to the state of the art.
- The authors should mention specific directions of future work related to potential enhancements of their proposed method.
Other comments/Typos
- Some comments are not properly defined in the Abstract.
- Citing references should be avoided in titles of subsections (Section 2)
- There are some incomplete references throughout the text, e.g., Page 11, line 398 “Table x”, Page 12, line 433, “related study[xx]”.
- In page 7, title: “3.2. UAS 3-layer model in CPS perspective” has been used twice.
Author Response
Please see attached PDF file with this response.
1. We mention the elliptic curve integrated encryption technique in a related work, and further discuss its overhead in 5.3.1. And sROS is also mentioned in a related work.
2. Section 5.3 emphasized that the proposed framework is appropriate for UAVs compared to the research mentioned in Section 4.
3. Our proposed framework ensures safe flight from false data injection by ensuring integrity. DOS attacks are not prepared. We refer to this as a future work in Conclude.
4. The second experimental graph showed Rosbridge characteristics by lowering its altitude using a regular client, but was changed to a constantly hovering experimental graph at 2 m to avoid misunderstanding.
5.Since it is impossible to implement existing studied systems and compare them experimentally in Section 6, we compare them separately in 5.3.
+ We modified ambiguous and unnatural expressions.

Reviewer 4 Report
The authors have investigated a security framework for Unmanned Aerial systems (UAS). The paper addresses an important topic, which is receiving much attention recently. However, I have the following comments:
- What are the chances that the digests can be modified by the attackers? Can the attacker inject false data in the digest to mislead the access control?
- The results analysis part needs to be improved. More attack scenarios need to be investigated in more experiments and the security metrics (on average over multiple experiments) need to be computed. Also, comparison with existing algorithms should be shown in terms of computational and security performances.
- UAS is not defined in the abstract, please define it.
Author Response
Please see attached PDF file with this response.
1. There is currently a hash algorithm that can crack. If it is used, an attacker can break the system. However, the SHA-256 algorithm used in experiments is impossible to break while operating the UAS. Even if an attacker takes the time to find out the hash, it is also impossible for the actual system to fail because of the short cycle in which the "key k" is exchanged. We added this content to 5.2.
2. Our proposed framework ensures safe flight from false data injection by ensuring integrity. Not ready for other attacks. We refer to this as a future work in the conclusion.
3. Several systems mentioned in Section 4 are covered in Section 5.3. New additions to Section 4 were also included in Section 5.3 and compared.
+ We modified ambiguous and unnatural expressions.

Round 2
Reviewer 1 Report
The authors still use the term 'stability', while they seem to mean 'security' (lines 55, 581), i.e. the original ROS is not protected against hacking, while their proposal remedies certain vulnerabilities to cyberattacks and thus enhances the [cyber] security of UAS. This is clearly different for the stability of the UAV or its trajectory when controlled from the ground.
Language must be improved to correct or make more clear statements like "DDS 157 requires several security requirements" (line 158), "Roland Dóczi et al. [23] claims" (line 257), "UAV has built an experimental environment" (line 552), "We explain the vulnerability of UAS with offboard ..." (line 587), etc.
I am not sure whether the authors have the right to use the image in Figure 1.
Author Response
Please see attached PDF file with this response.
1, 2. We agree with the reviewer's suggestion. We have revised the manuscript accordingly. And we have improved on other unclear sentences. The modifications are given in blue.
3. This photo is licensed under CC BY 4.0. Therefore, we marked the source of the photo in the references.

Reviewer 2 Report
Authors have made an effort to address reviewers comments. As as result, it has clearly improved, both in clarity and quality.
Author Response
Please see attached PDF file with this response.
We thank you to your time and considering on our submission.
We have further improved unnatural expressions. The modifications are available in blue.

Reviewer 3 Report
The authors have addressed my comments raised during the previous review round.
Author Response

(The authors gave the same response as above.)

Reviewer 4 Report
The authors have addressed my comments adequately.
Author Response

(The authors gave the same response as above.)
